# Third-Generation Antipsychotics and Lurasidone in the Treatment of Substance-Induced Psychoses: A Narrative Review

**DOI:** 10.3390/healthcare12030339

**Published:** 2024-01-29

**Authors:** Valerio Ricci, Domenico De Berardis, Giuseppe Maina

**Affiliations:** 1San Luigi Gonzaga Hospital, University of Turin, Regione Gonzole 10, 10043 Orbassano, Italy; giuseppe.maina@unito.it; 2NHS, Department of Mental Health, Psychiatric Service for Diagnosis and Treatment, Hospital “G. Mazzini”, ASL 4, 64100 Teramo, Italy; domenico.deberardis@aslteramo.it; 3Department of Neurosciences “Rita Levi Montalcini”, University of Turin, 10124 Torino, Italy

**Keywords:** substance-induced psychosis, psychostimulants, schizophrenia, third-generation antipsychotics, lurasidone, cariprazine, brexpiprazole, aripiprazole

## Abstract

This narrative review explores the efficacy and tolerability of third-generation antipsychotics (TGAs)—aripiprazole, cariprazine, brexpiprazole, and lurasidone—for the management of substance-induced psychosis (SIP). SIP is a psychiatric condition triggered by substance misuse or withdrawal, characterized by unique features distinct from those of primary psychotic disorders. These distinctive features include a heightened prevalence of positive symptoms, such as hallucinations and delusions, in addition to a spectrum of mood and cognitive disturbances. This review comprehensively investigates various substances, such as cannabinoids, cocaine, amphetamines, and LSD, which exhibit a greater propensity for inducing psychosis. TGAs exhibit substantial promise in addressing both psychotic symptoms and issues related to substance misuse. This review elucidates the distinctive pharmacological properties of each TGA, their intricate interactions with neurotransmitters, and their potential utility in the treatment of SIP. We advocate for further research to delineate the long-term effects of TGAs in this context and underscore the necessity for adopting an integrated approach that combines pharmacological and psychological interventions. Our findings underscore the intricate and multifaceted nature of treating SIP, highlighting the potential role of TGAs within therapeutic strategies.

## 1. Introduction

Based on the existing body of literature, there is robust evidence supporting a correlation between substance abuse and the initiation of psychotic symptoms. Numerous research findings indicate that illicit substances, like cannabinoids, cocaine, amphetamines, and hallucinogens, exhibit psychotomimetic properties [1,2]. This implies that their usage not only triggers temporary psychotic symptoms during acute intoxication but also may result in a syndrome closely resembling a primary psychotic disorder. In recent decades, a diverse range of novel psychoactive substances has emerged, encompassing synthetic cannabinoids, cathinone derivatives, psychedelic phenethylamines, new stimulants, synthetic opioids, tryptamine derivatives, phencyclidine-like dissociatives, piperazines, and GABAA/B receptor agonists. These substances are increasingly prevalent in the landscape of substance abuse [3].

A complex clinical challenge revolves around accurately differentiating substance-induced psychosis from a primary psychotic disorder or a psychotic disorder co-occurring with substance use. This poses a nuanced dilemma and an opportunity for thorough investigation, gaining significance when determining the optimal therapeutic approach for patients. The Diagnostic and Statistical Manual of Mental Disorders, Fifth Edition (DSM-5) [4] more accurately describes the substance/medication-induced psychotic disorder as a psychiatric condition characterized by delusions and/or hallucinations that arise during or shortly following the intoxication or withdrawal from a substance.

Additionally, the symptoms of a non-substance-induced psychotic disorder are not yet fully understood. Nevertheless, in this context, a particular definition has been formulated utilizing the diagnostic classification of “substance-related exogenous psychosis (SREP)”. This concept refers to a range of psychotic symptoms that are temporary or enduring and are associated with substance use. These include changes in consciousness; feelings of being persecuted; disorders affecting sensory perception, like visual and bodily hallucinations; impulsive behavior, self- or other-directed aggression, and psychomotor restlessness; fluctuations in mood; negative affects, such as indifference, lack of motivation, and inability to feel pleasure; an overwhelming feeling of being detached from reality; and maintained self-awareness [1]. Various clinical subtypes of substance-induced psychosis have been delineated and categorized based on prevalent symptoms and implicated neurotransmitters. An example is synthetic psychosis characterized by predominant dissociative reactions triggered by substances that impact glutamatergic pathways; this form of synthetic psychosis is characterized by dominant paranoia and auditory hallucinations resulting from substances that primarily target dopaminergic pathways. Additionally, there is a type of synthetic psychosis with prominent hallucinatory symptoms caused by substances that chiefly affect serotoninergic pathways, among other variations [5].

The occurrence of drug-induced psychosis appears to be associated with various pathogenetic mechanisms: (a) elevated levels of central dopamine, particularly for hallucinogens or psychedelic substances, stimulants, and cathinone derivatives; (b) activity as an agonist at cannabinoid CB1 receptors, particularly in substances related to cannabis; (c) agonist activity at 5-HT_2A_ receptors in hallucinogenic plants, newer phenethylamines, and tryptamine derivatives; (d) activity as an antagonist at NMDA receptors (n-methyl-D-aspartate receptors), seen in substances like ketamine and methoxetamine; and (e) activation of k-opioid receptors in plants, such as *Salvia divinorum* [6].

However, currently, there are no comprehensive guidelines for the management of these patients [7,8,9,10,11]. Antipsychotic agents are established as the primary modality for treating schizophrenia and are generally effective in managing substance-induced psychoses as well. These medications are categorized into two groups: typical antipsychotics, also referred to as first-generation antipsychotics (FGAs), and atypical antipsychotics, also known as second-generation antipsychotics (SGAs). The FGA class exhibits a relatively uniform pharmacological profile, whereas the SGA category displays a more diverse range, both pharmacologically and clinically. FGAs are marked by their strong antagonism of the dopamine D_2_ receptor. This action is beneficial in attenuating the heightened activity of the mesolimbic dopaminergic pathway, thereby mitigating hallucinations and delusions associated with this hyperactivity. However, D_2_ antagonism in the mesocortical pathway can exacerbate negative and cognitive symptoms. Additionally, a pronounced D_2_ blockade is linked to notable adverse effects, including extrapyramidal symptoms and hyperprolactinemia, arising from receptor inhibition in the nigrostriatal and tuberoinfundibular pathways. SGAs were developed with dual objectives: to reduce the incidence of these adverse effects by moderating the hyperactivity of the mesolimbic dopaminergic circuit in a more physiological manner and to address negative symptoms and cognitive deficits that were either unimproved or worsened by FGAs. Although SGAs lack a unifying pharmacological characteristic, a distinguishing feature of these drugs, in contrast to FGAs, is their concurrent antagonism of both dopamine D_2_ and serotonin 5-HT_2A_ receptors. The affinity ratio for these receptors has been considered as an indicative measure of their “atypicality”. Regarding substance-induced psychosis, clear indications have not emerged, as most cases studies refer to specific psychiatric disorders or the concurrent presence of withdrawal symptoms without specifically focusing on psychotic symptoms [12,13,14,15,16,17,18,19]. According to an Italian study based on a survey of antipsychotic drug prescription practices, in managing patients with psychosis induced by substances, particularly in acute situations, haloperidol was the most frequently chosen initial treatment drug, followed by aripiprazole and olanzapine, while in the maintenance phase, aripiprazole was the most-used first-choice drug, followed by olanzapine. Almost half the surveyed specialists used long-acting agents, while about a third did not. For clinicians prescribing long-acting medications, factors such as effectiveness, management of impulsivity, and the ability to target specific symptoms were key considerations in their decision-making process. During the maintenance phase, the emphasis was on enhanced patient adherence and drug tolerability [20]. The employment of long-acting antipsychotics as the initial treatment for patients experiencing their first episode of psychosis, alongside a substance use disorder, has been instrumental in lowering the chances of relapse and the need for rehospitalization, even in patients presenting with challenging prognostic factors [21].

In recent years, the development of molecules that act as partial agonists, instead of antagonists, at dopaminergic receptors has constituted a major breakthrough in the treatment of psychosis. This innovation has led to the introduction of third-generation antipsychotics, which include medications such as cariprazine, brexpiprazole, and aripiprazole. Additionally, lurasidone, though not classified within this newer generation, demonstrates distinct pharmacodynamic properties. It exhibits antagonistic effects on D_2_ dopaminergic receptors and serotonin 5-HT_2A_ and 5-HT_7_ receptors, and it is notable for its robust safety profile. These drugs are increasingly acknowledged for their therapeutic efficacy, safety, and ease of management in clinical settings.

### Research Goals and Objectives

The purpose of this review is to analyze in a narrative way the currently published literature on the long-term management of substance-induced psychosis, particularly evaluating the use of third-generation antipsychotics (TGAs) (aripiprazole, cariprazine, and brexpiprazole) and lurasidone, and to describe any recorded efficacy/tolerability issues. This research aims to fill a critical gap in the existing literature by providing a comprehensive overview of the effectiveness and challenges associated with TGAs in the management of substance-induced psychosis.

## 2. Materials and Methods

### 2.1. Data Extraction

A literature search was performed using PubMed (via MEDLINE) and Scopus databases on 21 December 2023, starting from January 1985. We used the following search string: (aripiprazole OR brexpiprazole OR cariprazine OR lurasidone) AND (psychosis OR schizophrenia OR schizoaffective) AND (“substance use disorder” OR cocaine OR alcohol OR cannabis OR heroin OR amphetamine OR methamphetamine OR psychostimulants OR “double diagnosis” OR “dual diagnosis”). During the search, the terms were also converted to (cannabis/THC/marijuana* OR; amphetamine METH OR * OR psychosis/non-affective psychosis/schizophrenia*). Only original articles related to third-generation antipsychotics (aripiprazole, brexpiprazole, and cariprazine) and lurasidone in the context of a substance-induced psychosis or comorbid schizophrenia and substance abuse and written in English were selected. Experimental and observational studies, post-marketing surveillance reports, case reports, case series, and fatality reports were included. The exclusion criteria included non-original research (e.g., reviews, commentaries, editorials, book chapters, and letters to the editor); non-full-text articles (e.g., meeting abstracts); and works in a language other than English. Research incorporating animal or in vitro experiments was considered for inclusion. Although letters to the editor, conference proceedings, and book chapters were not directly included in the literature review, they were referenced for obtaining additional secondary sources. Given the vast scope of the topic under study, employing a systematic method for data analysis and conducting statistical comparisons of the collected data were not feasible. The substances discussed exhibit distinct heterogeneity in their characteristics, which necessitated the use of a descriptive methodology to offer a comprehensive and detailed perspective on this subject.

### 2.2. Data Synthesis Strategy

Narrative reviews are commonly recognized for not incorporating databases and specific inclusion criteria [22,23]. Nevertheless, to enhance the clarity and transparency of our research, we have taken into consideration and systematized essential details, such as the authors’ names; year of publication; study design; demographic variables (gender, age, and psychiatric history); specifics of the antipsychotic drugs that were used (including the dosage and administration route); any concurrent substances; and the observed effects of these drugs on both psychotic symptoms and substance-related outcomes. Furthermore, we have considered animal studies and their possible impacts on future applications in humans. The effectiveness of a narrative review can be enhanced by incorporating methodological aspects from systematic reviews, which are designed to minimize bias in choosing articles, and using a robust bibliographic search technique. The literature search and selection process were conducted by two investigators (V.R. and D.D.B.) under the guidance of a supervisor, G.M. This process involved a two-stage independent review by the investigators, followed by a collaborative cross-check to ensure consistency and thoroughness.

Our initial investigation began with a review of titles and abstracts, progressing to a comprehensive analysis of full texts for articles potentially relevant to our study. Our goal was to encompass a wide range of literature on the subject. We identified 510 articles (143 from PubMed and 367 from Scopus) and selected 23 that corresponded with our predefined topics (12 on aripiprazole, 7 on cariprazine, and 2 each on lurasidone and brexpiprazole). The exclusion of the other 487 articles was due to factors such as irrelevance to the topic (299 articles) or non-compliance with our selection criteria, which included reviews, letters to the editor, commentaries, book chapters, non-English papers, and duplicates (188 articles), as shown in the flow diagram (Figure 1). We have arranged and presented our findings according to study type and substances analyzed in Appendix A (Appendix A). Characteristics of antipsychotic drugs are summarized in Appendix A. These formats offer a clear, detailed, and comprehensive view of the data, consistent with the descriptive approach of our review.

## 3. Results

### 3.1. Aripiprazole

Aripiprazole, classified as a partial agonist within the antipsychotic drug category, exhibits lower intrinsic activity at receptors compared to full agonists [24]. Its partial agonism at dopamine D_2_ receptors leads to (1) functional antagonism in the mesolimbic dopamine pathway, reducing positive symptoms caused by excessive dopamine activity, and (2) agonist activity in the mesocortical pathway, addressing negative symptoms and cognitive impairment due to reduced dopamine activity [25,26,27]. This dual effect contributes to significant improvements in both positive and negative psychotic symptoms.

Moreover, aripiprazole avoids the complete blockade of the nigrostriatal or tuberoinfundibular pathways, thereby preserving the avoidance of extrapyramidal symptoms and hyperprolactinemia [24]. It also demonstrates a favorable cardiac safety profile, preventing QTc prolongation and causing minimal weight gain or sedation. Focusing on substance use, aripiprazole shows promise in animal models, where acute administration prevented increased locomotion induced by stimulants, like amphetamine, cocaine, and methylphenidate, and attenuated their reinforcing properties without interfering with spontaneous motor activity [26]. It reversed amphetamine-associated anhedonia and prevented the reinstatement of cocaine-seeking behavior [27], suggesting its potential in alleviating withdrawal symptoms linked to dopamine depletion and maintaining balanced dopamine neurotransmission in drug-dependent behavior [28,29,30].

Clinical observations further support these findings. A 22-year-old male patient meeting ultra-high-risk criteria for psychosis benefited from 10 mg of aripiprazole, though without a decrease in smoking frequency [31]. Another case highlighted aripiprazole’s role in reducing cannabis intake in a schizophrenic patient [32]. Thurston and colleagues observed that adolescents hospitalized for co-occurring psychosis and cannabis use disorder showed a rapid reduction in acute psychotic symptoms with aripiprazole compared to risperidone [33]. In trials comparing aripiprazole with other antipsychotics, it demonstrated superior efficacy in reducing negative symptoms of amphetamine-induced psychosis compared to risperidone, which was more effective against positive symptoms [34,35]. A randomized study found aripiprazole comparable to quetiapine in alleviating psychotic symptoms but more effective in reducing cocaine dependence and usage [36]. Beresford’s study indicated its potential role in lowering both the desire and use of cocaine in schizophrenic patients [37].

However, not all studies yielded positive outcomes. Aripiprazole was no more effective than a placebo in maintaining abstinence from methamphetamine use but did facilitate treatment retention and reduce the severity of psychotic symptoms [38].

### 3.2. Cariprazine

Cariprazine, a third-generation antipsychotic drug approved for schizophrenia [39], is notable for its partial agonism toward D_2_ receptors. Distinguishing itself from other second-generation antipsychotics (SGAs), it demonstrates a lower intrinsic activity at the D_2_ receptor compared to aripiprazole and brexpiprazole, approximately 0.15. This positioning suggests a minimized capacity to activate the D_2_ receptor, offering an intermediary profile between aripiprazole and traditional receptor antagonists with zero intrinsic activity [39,40,41,42]. The reduced intrinsic activity of cariprazine is associated with a decreased likelihood of side effects, such as restlessness and akathisia, commonly observed with aripiprazole [40].

Cariprazine’s primary distinction lies in its high affinity for the dopaminergic D_3_ receptor, exceeding that of endogenous dopamine. This high D_3_ affinity is significant, considering that most antipsychotics have a lower affinity for this receptor, leading to inadequate D_3_ receptor occupancy in the brain. This unique affinity is crucial for cariprazine’s clinical efficacy, as indicated by PET studies on schizophrenic patients, demonstrating a stronger effect on the D_3_ receptor compared to the D_2_ receptor [41,42,43,44,45,46,47] (see Appendix A).

Cariprazine also exhibits significant serotonergic activity, with high affinity for the 5-HT2B receptor and moderate affinity for the 5-HT_2A_ and 5-HT_1A_ receptors. This receptor profile is somewhat distinct from those of many SGAs, which tend to have high affinities for the 5-HT_2A_ receptor. The 5-HT_2A_/D_2_ affinity ratio of cariprazine is lower than those of other antipsychotic drugs. Additionally, it has low affinity for the 5-HT7 and 5-HT2C receptors as well as the noradrenaline α1A and α1C receptors. Its interactions with the 5-HT6, α1a, and α2b receptors and other potential targets are minimal in therapeutic activity [42,43].

When comparing cariprazine with other partial agonists, such as blonanserin [48,49], an antagonist of dopaminergic D_2_ and D_3_ receptors, its D_3_/D_2_ affinity ratio is notably higher. This suggests a potential influence on negative symptoms, a theory supported by its unique receptor action. In contrast, brexpiprazole shows higher 5-HT_2A_/D_2_ and 5-HT_1A_/D_2_ ratios. The combined dopaminergic and serotonergic receptor activities of these drugs modulate specific brain circuits and neurotransmitter release, as observed in increased dopamine release in the prefrontal cortex with olanzapine and lurasidone, enhancing cognitive functions [50]. Similarly, cariprazine affects neurotransmitter release in the nucleus accumbens and hippocampus, mainly via D_3_ receptor interaction, influencing levels of dopamine, norepinephrine, serotonin, and glutamate. This effect is comparable to that of dopaminergic D_3_ antagonists, contributing to cariprazine’s therapeutic impact [50].

In conclusion, the receptor profile of cariprazine provides significant insights into cariprazine’s mechanism of action and ability to modulate various neurotransmitter systems. Unlike other antipsychotics, cariprazine’s efficacy is based on not only its receptor profile but also its capacity to modulate intracellular mechanisms downstream of these receptors [51,52]. Its dual action as a partial agonist at both D_2_ and D_3_ receptors, with a stronger effect on the latter, is particularly effective in improving negative symptoms, setting it apart from other SGAs [53].

Another intriguing facet of cariprazine pertains to its activity in the context of substance abuse. Experimental studies have illustrated cariprazine’s capacity to mitigate the stimulating effects of cocaine and forestall relapses associated with the abused substance. This observed activity seems to correlate with its partial agonism for dopamine receptors D_2_ and D_3_ [54]. Notably, a recent study has demonstrated comparable anti-abuse effects using (±)VK4-40, a novel selective partial agonist for the dopaminergic receptor D_3_. This finding suggests the potential contribution of this specific receptor component to the action of cariprazine [55]. Rodriguez et al. [56] presented evidence of the positive impact of cariprazine on positive, negative, and cognitive symptoms in a schizophrenic patient with an extensive history of substance abuse. The patient underwent a transition from haloperidol to cariprazine, leading to the comprehensive amelioration of symptoms associated with schizophrenia. Ricci and colleagues [57] illustrated the favorable effect of cariprazine on mitigating psychotic symptoms induced by methamphetamine in a 25-year-old male who had shown only partial responsiveness to olanzapine and risperidone. Concerning methamphetamine-induced psychosis, Truong et al. [58] showcased both the antipsychotic and anticraving effects of cariprazine through two case reports featuring men aged 33 and 51. These individuals not only presented significant positive symptoms but also demonstrated pronounced substance-seeking behavior. In both cases, cariprazine was initiated at dosages of 1.5 mg and 3 mg, respectively, following prior use of olanzapine and paliperidone, which produced limited efficacy and resulted in significant side effects, such as weight gain and metabolic imbalance. In a separate case report [59] the administration of 3 mg of cariprazine effectively alleviated positive symptoms in a 31-year-old male who was HIV positive and had a history of abusing methamphetamine, mephedrone, cocaine, and alcohol.

Another case report [60] detailed symptomatic improvement in a patient with bipolar disorder and a history of poly-substance abuse, including cannabis, LSD, and methamphetamine. In this instance, cariprazine was introduced as an adjunct to the ongoing therapy (valproate, SSRI, and bupropion) at a maximum dosage of 3 mg. This intervention led to the alleviation of substance cravings and enhancement of affective symptoms.

Gentile and colleagues [61] outlined a case involving a 23-year-old patient diagnosed with schizophrenia spectrum disorder and a psychotic onset induced by extensive cannabis use. Following the discontinuation of lurasidone owing to ineffectiveness, the administration of cariprazine not only led to an improvement in both negative and positive symptoms but also resulted in a reduction in cannabinoid intake.

### 3.3. Brexpiprazole

Brexpiprazole, an atypical antipsychotic, received FDA approval in July 2015 for treating schizophrenia and as an adjunct therapy for managing major depressive disorder (MDD). Acting as a partial agonist at 5-HT_1A_ and D_2_ neuroreceptors, brexpiprazole also engages with noradrenergic receptors, although the clinical significance of this interaction is not fully understood. The safety and efficacy of brexpiprazole were evaluated in four finished placebo-controlled Phase III trials—two targeting major depressive disorder (MDD) as an add-on to antidepressants and two for schizophrenia. These studies showed that brexpiprazole was more effective than a placebo at certain dosages for both disorders [62,63,64].

For schizophrenia, the recommended initiation is 1 mg once daily, titrating to a target dose from 2 mg to 4 mg daily. As an adjunct therapy for MDD, the initiation dose is advised at 0.5 mg or 1 mg once daily, with a weekly increase to a target dose of 2 mg. Contraindications include prior hypersensitivity reactions to similar medications. Common adverse reactions involve weight gain and akathisia, with additional associations with metabolic changes, like dyslipidemia and hyperglycemia. Precautions and warnings for this medication include cerebrovascular adverse reactions in elderly patients with dementia-related psychosis, the risk of neuroleptic malignant syndrome, tardive dyskinesia, leukopenia, orthostatic hypotension, and seizures. A prominent black box warning highlights the increased risk of mortality in elderly patients with dementia-related psychosis, as well as the potential for suicidal thoughts and behaviors in children, adolescents, and young adults. Additionally, its use during pregnancy is cautioned against owing to the risk of extrapyramidal and/or withdrawal symptoms in neonates exposed during the third trimester. Brexpiprazole is metabolized by hepatic enzymes, and dose adjustments are necessary if the CYP2D6 or CYP3A4 superfamilies of enzymes are impacted [65]. Brexpiprazole functions as a partial agonist at dopamine D_2_ and serotonin 5-HT_1A_ receptors and a potent antagonist at serotonin 5-HT_2A_, α_1B_, and α_2C_ adrenergic receptors. In contrast to aripiprazole, brexpiprazole exhibits significantly greater potency at these three receptors: 5-HT_2A_, 5HT_1A_, and α_1B_. Despite commonly reported side effects, such as EPS and akathisia, the strong affinity of brexpiprazole to these receptors may contribute to a reduction in the occurrence of these symptoms. Brexpiprazole exhibits a higher intrinsic activity at the serotonin 5-HT_2A_ receptor and a lower intrinsic activity at the dopamine D_2_ receptor along with a stronger affinity for the norepinephrine transporter [66]. Hyperprolactinemia, an often-undesirable side effect of many antipsychotics, is largely due to the blockade of dopamine D_2_ receptors, which interrupts the dopaminergic inhibition of the prolactin release. In a detailed evaluation, brexpiprazole shows moderate antagonist activity at dopamine D_3_ and serotonin 5-HT_2B_ and 5-HT_7_ receptors, as well as α_1A_ and α_1D_ receptors. It also has moderate affinity for histamine H_1_ receptors and low affinity for muscarinic cholinergic M_1_ receptors. This pharmacodynamic profile not only enhances brexpiprazole’s efficacy but also positions brexpiprazole as a potentially preferred alternative based on patient tolerability and therapy goals [67].

Brexpiprazole undergoes primary metabolism by the enzymes CYP3A4 and CYP2D6. The major metabolite, DM-3411, constitutes between 23% and 48% of brexpiprazole’s exposure at a steady state, although it has not been demonstrated to contribute to any antipsychotic effects. Various factors influence the rate of brexpiprazole metabolism, subsequently impacting its overall exposure (AUC). Patients who use strong CYP3A4 inhibitors (like erythromycin or itraconazole) or potent CYP2D6 inhibitors (such as bupropion, fluoxetine, or paroxetine) may experience increased exposure to the drug. This heightened exposure is also seen in individuals who are poor metabolizers of CYP2D6. On the other hand, using potent CYP2D6 inducers (for instance, rifampicin or glucocorticoids) is likely to decrease the exposure to brexpiprazole. Additionally, patients with moderate to severe liver impairment (classified as Child–Pugh Class B or C) or those with moderate to severe kidney impairment (with a creatinine clearance rate below 60 mL/min) are subject to increased drug exposure, which may require adjustments in dosage [68,69].

Brexpiprazole demonstrates efficacy in acute schizophrenia according to two studies [70,71]. Although the 2 mg dose showed inconsistent results, the recommended 4 mg dosage exhibited more consistent benefits. Higher brexpiprazole dosages, especially at 4 mg, significantly improved the PANSS-EC score and PANSS scores for negative symptoms, disorganized thought, and uncontrolled hostility/excitement. In a 52-week maintenance study, brexpiprazole outperformed the placebo, prolonging the time to exacerbation and showing significant benefits in psychosocial, occupational, and cognitive functioning, including attention/vigilance and visual learning [63]. Brexpiprazole also demonstrated cognitive improvement in animal models, distinguishing itself from aripiprazole in this regard. Short-term trials showed a good safety profile, with weight gain being the only common adverse event, and long-term studies indicated a decrease in the mean bodyweight change. Akathisia was dose-dependent but generally mild, with no treatment discontinuations [72]. Other adverse effects were comparable to those of the placebo, and minimal impacts on glucose, lipids, prolactin, and the QTc interval were observed.

As of the current state, only two studies shed light on the role of brexpiprazole in substance-induced psychoses [73,74]. Nickols and colleagues conducted research to explore its effects in a mouse model of opioid dependence, providing preclinical evidence for the efficacy of brexpiprazole as a modulator of dopamine-dependent behaviors during opioid use and withdrawal. This research contributes valuable insights into the potential application of brexpiprazole in forthcoming human studies. The findings suggest a plausible role as a pharmacological adjunct not only in addressing the emergence of substance-induced psychoses but also in mitigating craving phenomena [73].

One additional study highlights the role of brexpiprazole in substance-induced psychoses. Specifically, Kung and colleagues demonstrate the effectiveness of brexpiprazole in treating psychosis during cannabis withdrawal in a 25-year-old individual at a daily dosage of 3 mg [74].

### 3.4. Lurasidone

Lurasidone, like other second-generation antipsychotics, functions as a complete antagonist at dopamine D_2_ and serotonin 5-HT_2A_ receptors, with binding affinities (Ki) of 1 nM and 0.5 nM, respectively. A distinctive feature of lurasidone is its high affinity for serotonin 5-HT_7_ receptors (0.5 nM, on par with its affinity for dopamine D_2_ and 5-HT_2A_ receptors) and its partial agonist activity at 5-HT_1A_ receptors (Ki, 6.4 nM). The serotonin 5-HT_7_ receptor is of significant interest as it is linked to potential procognitive and antidepressant effects. The 5-HT_1A_ receptor is considered important in the treatment of major depressive disorder and schizophrenia. Notably, lurasidone lacks affinity for histamine H_1_ and muscarinic M_1_ receptors, contributing to its characteristics of low sedation, minimal weight gain, and limited interference with cognitive and functional assessments [75,76]. The pharmacokinetic characteristics of lurasidone support its suitability for once-daily administration, as it has an elimination half-life of 18 h [77,78]. When lurasidone was administered with food, both the mean Cmax and the area under the curve were approximately threefold and twofold greater, respectively, compared to those for administering lurasidone while fasting [79]. Based on these findings and clinical trial results, it is recommended to take lurasidone once daily in the evening, either with a meal or within 30 min after eating. Notably, lurasidone absorption remains unaffected by the fat content of the ingested food [79]. Lurasidone is primarily metabolized by the CYP3A4 enzyme system. Consequently, its use is contraindicated when there are strong inducers or inhibitors of CYP3A4 present. Within the category of psychotropic medications, notable examples of strong inhibitors of CYP3A4 include fluvoxamine and fluoxetine. On the other hand, a well-known strong inducer of CYP3A4 is carbamazepine. When moderate inhibitors of CYP3A4 are present, the suggested initial dose of lurasidone is 20 mg/day instead of 40 mg/day, and the highest recommended dose is 80 mg/day rather than 160 mg/day. Lurasidone’s pharmacokinetics do not interfere with those of other drugs, including lithium, valproate, or those metabolized by the CYP3A4 pathway [80].

In patients who have moderate or severe renal or hepatic impairment, the advised initial dosage of this medication is set at 20 mg per day. For those with moderate-to-severe renal impairment or moderate hepatic impairment, the maximum dosage should not surpass 80 mg per day. In cases of severe hepatic impairment, the dosage should be restricted to a maximum of 40 mg per day. Lurasidone demonstrated good tolerability, with consistent side effects observed in both short-term and long-term use. In trials spanning six weeks, the most frequently reported adverse reactions to lurasidone included drowsiness, restlessness, nausea, Parkinson-like symptoms, and sleeplessness [81]. Finally, lurasidone was associated with less weight gain and fewer metabolic disturbances than brexpiprazole [82,83].

An analysis that combined data from eight short-term (6-week) placebo-controlled studies conducted across the United States, Europe, Asia, and South America revealed the effectiveness of lurasidone (40–160 mg/day) in treating schizophrenia [84]. The findings indicated that lurasidone demonstrated efficacy compared to a placebo, leading to improvements in positive symptoms, negative symptoms, and general psychopathology. Additionally, the meta-analysis highlighted that lurasidone was well-tolerated, showing minimal impacts on bodyweight, glucose, and lipid parameters. Continuation studies spanning from 6 to 22 months affirmed the sustained efficacy of lurasidone in the treatment of schizophrenia, maintaining minimal effects on bodyweight and metabolic parameters [85,86,87,88]. A recent 26-week open-label study, focusing on lurasidone at doses of 40–80 mg/day, extended these positive results to patients with schizophrenia in Asia, including Japan, Taiwan, Korea, and Malaysia [89].

Concerning the use of lurasidone for psychopathological conditions induced by substances, there is limited literature available. Despite being an approved medication for psychotic disorders in individuals aged 15 and older, there are currently no trials or extensive clinical observations of lurasidone in the context of substance-induced psychoses. Only two studies have addressed the role of lurasidone in these circumstances.

The initial study [90] comprises four clinical observations of young patients experiencing cannabis-induced psychosis. These individuals exhibited improvements in both positive and negative symptomatologies as well as mood, following the administration of lurasidone. The dosage of lurasidone ranged from 74 mg to 111 mg and was prescribed as both an initial treatment in drug-naive patients and after the failure of therapies with aripiprazole and paliperidone. In contrast, the second study [91] appears to be more comprehensive, focusing on the utilization of lurasidone in young individuals with complex psychopathological conditions. Notably, the study reports the beneficial use of lurasidone in a 14-year-old with a history of alcohol, cannabis, and LSD abuse, coupled with behavioral issues (self-injurious behaviors) and psychotic symptoms, such as auditory hallucinations.

## 4. Discussion

From the above-mentioned data, third-generation antipsychotics and lurasidone emerge as promising therapeutic strategies in the treatment of substance-induced psychoses. Aripiprazole has been effective in improving a wide range of psychotic symptoms, including both positive and negative aspects, as well as impacting substance use disorders positively. Its mode of action is unique; it is a partial agonist, meaning it does not stimulate receptors as strongly as full agonists. Its effectiveness lies in its dual role: it diminishes positive symptoms by antagonizing the mesolimbic dopamine pathway and improves negative symptoms and cognitive deficits by activating the mesocortical pathway. This selective mechanism helps aripiprazole to avoid severe side effects, like motor disorders and elevated prolactin levels, that are common for other antipsychotics. Additionally, it is known for its cardiac safety, causing negligible QTc prolongation and having a low risk of weight gain or sedation. In research with animals, aripiprazole has been observed to curb the heightened activity caused by stimulants, such as amphetamine, cocaine, and methylphenidate, and reduce their addictive qualities without hampering normal motor functions. It also reverses the lack of pleasure associated with amphetamine use and hinders the recurrence of cocaine-seeking behaviors. These findings indicate that aripiprazole might be effective in easing withdrawal symptoms associated with dopamine deficiency and, owing to its broad receptor activity, could represent a new strategy for achieving balanced dopamine levels in the treatment of drug addiction.

Cariprazine’s receptor profile, particularly its high affinity for the D_3_ receptor and reduced intrinsic activity at D_2_ receptors, makes it an effective treatment for schizophrenia, improving both positive and negative symptoms. Its ability to modulate different neurotransmitter systems further highlights its potential as a distinct and effective antipsychotic medication. Cariprazine’s role in substance abuse treatment is noteworthy. From the studies that were reviewed, its effectiveness emerges in reducing the stimulating effects of substances, like cocaine, and mitigating cravings and relapses. This effect is possibly due to its partial agonism at D_2_ and D_3_ receptors. Case studies show cariprazine’s beneficial impact on schizophrenic patients with a history of substance abuse and its effectiveness in treating psychosis induced by substances, like methamphetamine.

Brexpiprazole exhibits potential efficacy in the domain of substance abuse therapy. Primarily indicated for the management of schizophrenia and as an adjunctive treatment in major depressive disorder (MDD), its unique pharmacodynamic properties extend to the mitigation of substance-induced psychotic disorders. Operating as a partial agonist at the 5-HT_1A_ and D_2_ neuroreceptors, brexpiprazole also engages with noradrenergic receptors. Consequently, albeit preliminary and limited in scope, research indicates its utility in addressing psychotic sequelae associated with cannabis consumption and in modulating dopaminergic activity in heroin-exposed rodents.

Lurasidone has shown potential in treating psychopathological conditions related to substance abuse, although research in this area is limited. Known for its antagonistic action at dopamine D_2_ and serotonin 5-HT_2A_ receptors and strong affinity for serotonin 5-HT_7_ receptors, lurasidone is unique in its partial agonism at 5-HT_1A_ receptors. This receptor profile contributes to its low sedative effects and minimal impact on weight and cognitive functions, making it an appealing option in treating substance-induced psychoses. In the context of substance abuse, recent studies have shown lurasidone to be effective in treating young individuals with substance-induced psychosis, particularly from cannabis, improving various symptoms, including mood. It has also been beneficial for a complex case involving a young person with alcohol, cannabis, and LSD abuse along with behavioral and psychotic symptoms. It is important to consider that lurasidone is an approved medication for treating schizophrenia in individuals as young as 13, an age group particularly susceptible to substance use. Consequently, this could make it a feasible option for treatment in this younger demographic in the future.

Finally, despite aripiprazole showing more scientific evidence, there are currently limited specific clinical studies demonstrating its long-term efficacy. Several critical issues have emerged from our observation, including the following:

First, these drugs may exhibit variable efficacies in managing psychotic symptoms associated with substance use, as the nature and complexity of substance-induced psychoses can differ significantly from those purely psychiatric in origin. Additionally, third-generation antipsychotics may not optimally address specific aspects of substance-induced psychoses, such as the management of cognitive disorders and compulsive impulses, which often characterize these conditions.

Another significant limitation concerns side effects. Third-generation antipsychotics, although to a lesser extent, can induce metabolic disorders and increase the risk of tardive dyskinesia, posing additional challenges in the treatment of substance-induced psychoses. Furthermore, patient compliance may become an issue due to these side effects, compromising treatment adherence.

## 5. Conclusions

In summary, the core feature of third-generation antipsychotics, like aripiprazole, brexpiprazole, and cariprazine, is their partial agonism at D_2_ dopaminergic receptors. This characteristic offers at least three potential benefits compared to the traditional antagonism approach:

-In cases of mesolimbic dopaminergic overactivity linked to positive symptoms, the partial agonist competes with dopamine for receptors. This competition displaces dopamine, decreasing the system’s excessive activity and returning it to a ‘physiological range’;-The effects of the partial agonist do not typically include significant adverse reactions, such as extrapyramidal symptoms or hyperprolactinemia. This is because the molecule’s intrinsic activity prevents a substantial reduction in dopaminergic functionality at the striatal and pituitary levels;-Owing to their intrinsic activities, partial agonists may bolster weakened dopaminergic transmission in the prefrontal cortex. This enhancement could improve cognitive dysfunctions and symptoms linked to diminished dopaminergic functionality. These pharmacodynamic properties make partial agonists especially useful in cases of substance-induced psychosis.

Concerning lurasidone, its efficacy lies in its binding to D_2_ receptors, which helps in reducing positive symptoms. Additionally, its agonist activity at the 5-HT_2A_ and 5HT_7_ receptors results in the release of dopamine, contributing to the improvement of affective and cognitive symptoms. This mechanism of action could provide a foundation for additional research and the initiation of studies on patients with substance-induced psychosis.

Finally, it is essential to consider that third-generation antipsychotics may not fully address the underlying cause of substance-induced psychoses, linked to the intake of specific psychoactive substances. Therefore, a comprehensive therapeutic approach should include not only symptomatic treatment with antipsychotics but also targeted intervention for substance dependence and relapse prevention through psychological interventions and rehabilitation programs. In summary, although third-generation antipsychotics are valuable in treating psychoses, their effectiveness and appropriateness in treating substance-induced psychoses require particular attention and an integrated approach.

## Figures and Tables

**Figure 1 healthcare-12-00339-f001:**
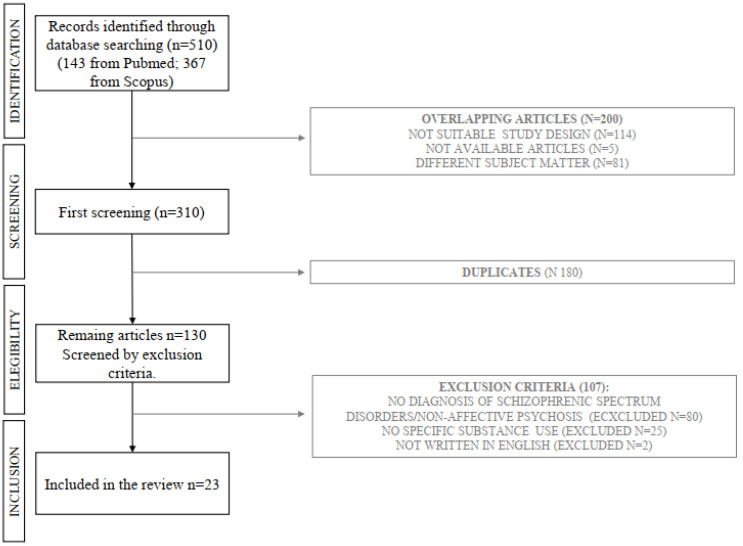
Flowchart of study search and selection processes.

## Data Availability

The data presented in this study are available on request from the corresponding author. The data are not publicly available owing to privacy reasons.

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
