# Peer review of "Third-Generation Antipsychotics and Lurasidone in the Treatment of Substance-Induced Psychoses: A Narrative Review"

_healthcare, 2024, doi:10.3390/healthcare12030339_

Round 1
Reviewer 1 Report
Comments and Suggestions for Authors
Dear authors, although I have greatly appreciated your work on narrative revision, the structural scaffolding of the manuscript has many shortcomings that make it not work well for the stated objectives. For the reasons I will state, the manuscript should be rejected but I appreciate the research and editing effort, and therefore I will only suggest major revisions:
1) there is too much emphasis in the abstract on the concept of "comprehensive" which for editorial balance should be removed;
2) more emphasis should be given in the introduction to the concept of psychosis and psychotic disorders, referring to the DSM-V-TR (2022) and not to the previous version as erroneously contained in the manuscript; moreover, the difference between first-, second, and third-generation drugs should be explained, and why this revision focuses only on the last category (not only as goals but as research centrality);
3) the section on research goals and objectives needs to be separated from the introduction by assigning a specific paragraph;
4) the materials and methods section needs to indicate the date of the start of the Pubmed search (e.g., January 1990) and needs to be supplemented with a PRISMA-type flowchart even though it has been declared a "narrative" review, as it is clear from lines 123-132 that it is a systematic review and the authors "used" the term "narrative" to avoid the PRISMA flowchart and registration with PROSPERO. This confusion of form is deceptive, and the authors are asked to decide whether: a) to leave "narrative" and include a flowchart that makes clear the rationale for article selection + a table that schematizes for all selected articles the key concept of the research; b) to proceed with the transformation to "systematic" and provide PRISMA+PROSPERO+table.
5) in the results section, the process of selection of those drugs is not clear, in the absence of at least 1 summary table with the technical characteristics of use and administration of each of them;
6) the discussion section is too sparse, and needs to be supplemented with a specific argumentation, drug by drug, of strengths and weaknesses, otherwise it is not even a review paper but a simple "editorial" + a summary table of discussion outcomes;
7) the conclusions are unimpressive, compared to the study topic and deserve more elaboration;
8) the notes must be revised based on the Journal's editorial rules, and at least 50% must be from the last five years.
Good work!
Author Response
Dear REVIEWER 1,
Dear authors, although I have greatly appreciated your work on narrative revision, the structural scaffolding of the manuscript has many shortcomings that make it not work well for the stated objectives. For the reasons I will state, the manuscript should be rejected but I appreciate the research and editing effort, and therefore I will only suggest major revisions:
Reply: Thank you for taking the time to provide a thorough review of our manuscript. We deeply appreciate your recognition of the research and editing efforts that went into our work. We also understand your concerns regarding the structural aspects of the manuscript and how they may not effectively meet our stated objectives. We are committed to addressing these shortcomings and are prepared to undertake major revisions to ensure that our manuscript not only meets but exceeds the standards expected by your journal.
-There is too much emphasis in the abstract on the concept of "comprehensive" which for editorial balance should be removed;
Reply: Thank you for your observation regarding the use of the term "comprehensive" in the abstract. In response to your comment and in line with the second reviewer, we changed the abstract as follows
” This narrative review delves into the efficacy and tolerability of third-generation antipsychotics (TGAs) – aripiprazole, cariprazine, brexpiprazole, and lurasidone – for the management of substance-induced psychosis (SIP). SIP is a psychiatric condition triggered by substance misuse or withdrawal, characterized by unique features distinct from primary psychotic disorders. These distinctive features include a heightened prevalence of positive symptoms, such as hallucinations and delusions, in addition to a spectrum of mood and cognitive disturbances. This review comprehensively investigates various substances, such as cannabinoids, cocaine, amphetamines, and LSD, which exhibit a greater propensity for inducing psychosis.TGAs exhibit substantial promise in addressing both psychotic symptoms and issues related to substance misuse. This review elucidates the distinctive pharmacological properties of each TGA, their intricate interactions with neurotransmitters, and their potential utility in the treatment of SIP. We advocate for further research to delineate the long-term effects of TGAs in this context and underscore the necessity of adopting an integrated approach that combines pharmacological and psychological interventions. Our findings underscore the intricate and multifaceted nature of treating SIP, highlighting the potential role of TGAs within therapeutic strategies.”
-more emphasis should be given in the introduction to the concept of psychosis and psychotic disorders, referring to the DSM-V-TR (2022) and not to the previous version as erroneously contained in the manuscript; moreover, the difference between first-, second, and third-generation drugs should be explained, and why this revision focuses only on the last category (not only as goals but as research centrality);
Reply: Thank you for this valuable suggestion. We acknowledge the error in citing the previous version of the DSM and will rectify this by referring to the latest DSM-V-TR (2022). This will ensure that our manuscript is aligned with the most current diagnostic criteria and conceptualizations of psychotic disorders. We modified as follows:
The Diagnostic and Statistical Manual of Mental Disorders, Fifth Edition (DSM-5) T.R. (4), more accurately describes the substance/medication-induced psychotic disorder as a psychiatric condition characterized by delusions and/or hallucinations that arise during or shortly following the intoxication or withdrawal from a substance.
- Inizio modulo
-the section on research goals and objectives needs to be separated from the introduction by assigning a specific paragraph
Reply: Thank you for this suggestion. We provided to separate the introduction from specific research goals
-the materials and methods section needs to indicate the date of the start of the Pubmed search (e.g., January 1990) and needs to be supplemented with a PRISMA-type flowchart even though it has been declared a "narrative" review, as it is clear from lines 123-132 that it is a systematic review and the authors "used" the term "narrative" to avoid the PRISMA flowchart and registration with PROSPERO. This confusion of form is deceptive, and the authors are asked to decide whether: a) to leave "narrative" and include a flowchart that makes clear the rationale for article selection + a table that schematizes for all selected articles the key concept of the research; b) to proceed with the transformation to "systematic" and provide PRISMA+PROSPERO+table.
Reply: We appreciate your in-depth and considerate feedback on the categorization and methodological approach of our review. It's important to note that our review is conceptualized as a narrative review. The chosen methodology and approach aim to offer an extensive and thematic survey of the existing literature, rather than a methodical compilation and evaluation of data. Nevertheless, we acknowledge the significance of explicitly stating our search criteria and selection justifications to bolster the review's reliability and utility. Following your suggestions, we have incorporated a flow chart and introduced an additional column in the tables for each examined drug, explicitly detailing the key concept. We changed Material and Method chapter as follows:
Narrative reviews are commonly recognized for not incorporating databases and specific inclusion criteria, as highlighted by studies like Cipriani and Geddes (2003) and Collins and Fauser (2005). Nevertheless, to enhance the clarity and transparency of our research, we have taken into consideration and systematized essential details such as the authors' names, year of publication, study design, demographic variables (gender, age, psychiatric history), specifics of the antipsychotic drugs used (including dosage and administration route), any concurrent substances, and the observed effects of these drugs on both psychotic symptoms and substance-related outcomes. Furthermore, we have considered animal studies and their possible impact on future applications in humans The effectiveness of a narrative review can be enhanced by incorporating methodological aspects from systematic reviews, which are designed to minimize bias in choosing articles, and by using a robust bibliographic search technique
-in the results section, the process of selection of those drugs is not clear, in the absence of at least 1 summary table with the technical characteristics of use and administration of each of them;
Reply: Thank you very much for this valuable comment: we added in supplementary file a Table 5 with the characteristics requested.
-the discussion section is too sparse, and needs to be supplemented with a specific argumentation, drug by drug, of strengths and weaknesses, otherwise it is not even a review paper but a simple "editorial" + a summary table of discussion outcomes
Reply: Thank you for your insightful feedback on the discussion section of our manuscript. In response to your suggestions, we will undertake the following revisions: we enrich the discussion section by providing a more comprehensive analysis of each drug covered in our review drawing on the latest research and clinical findings. This approach aims to provide a balanced view of each drug, considering factors such as efficacy, safety profile, pharmacokinetics, and patient tolerability. We changed the discussion as follows:
Aripiprazole has been effective in improving a wide range of psychotic symptoms, including both positive and negative aspects, as well as impacting substance use disorders positively. Its mode of action is unique; it's a partial agonist, meaning it doesn't stimulate receptors as strongly as full agonists. Its effectiveness lies in its dual role: it diminishes positive symptoms by antagonizing the mesolimbic dopamine pathway and improves negative symptoms and cognitive deficits by activating the mesocortical pathway. This selective mechanism helps aripiprazole to avoid severe side effects like motor disorders and elevated prolactin levels, common in other antipsychotics. Additionally, it's known for its cardiac safety, causing negligible QTc prolongation and having a low risk of weight gain or sedation. In research with animals, aripiprazole has been observed to curb the heightened activity caused by stimulants such as amphetamine, cocaine, and methylphenidate, and reduce their addictive qualities without hampering normal motor functions. It also reverses the lack of pleasure associated with amphetamine use and hinders the recurrence of cocaine-seeking behaviors. These findings indicate that aripiprazole might be effective in easing withdrawal symptoms associated with dopamine deficiency and, due to its broad receptor activity, could represent a new strategy for achieving balanced dopamine levels in the treatment of drug addiction.
Cariprazine’s receptor profile, particularly its high affinity for the D3 receptor and reduced intrinsic activity at D2 receptors, makes it an effective treatment for schizophrenia, improving both positive and negative symptoms. Its ability to modulate different neurotransmitter systems further highlights its potential as a distinct and effective antipsychotic medication. Cariprazine's role in substance abuse treatment is noteworthy. From the Studies reviewed, emerge its effectiveness in reducing the stimulating effects of substances like cocaine and mitigating cravings and relapses. This effect is possibly due to its partial agonism at D2 and D3 receptors. Case studies show cariprazine’s beneficial impact on schizophrenic patients with a history of substance abuse and its effectiveness in treating psychosis induced by substances like methamphetamine.
Brexpiprazole exhibits potential efficacy in the domain of substance abuse therapy. Primarily indicated for the management of schizophrenia and as an adjunctive treatment in Major Depressive Disorder (MDD), its unique pharmacodynamic properties extend to the mitigation of substance-induced psychotic disorders. Operating as a partial agonist at the 5-HT1A and D2 neuroreceptors, brexpiprazole also engages with noradrenergic receptors. Consequently, albeit preliminary and limited in scope, research indicates its utility in addressing psychotic sequelae associated with cannabis consumption and in modulating dopaminergic activity in heroin-exposed rodents
Inizio moduloLurasidone has shown potential in treating psychopathological conditions related to substance abuse, although research in this area is limited. Known for its antagonistic action at dopamine D2 and serotonin 5-HT2A receptors, and strong affinity for serotonin 5-HT7 receptors, lurasidone is unique in its partial agonism at 5-HT1A receptors. This receptor profile contributes to its low sedative effects and minimal impact on weight and cognitive functions, making it an appealing option in treating substance-induced psychoses. In the context of substance abuse recent studies has shown lurasidone to be effective in treating young individuals with substance-induced psychosis, particularly from cannabis, improving various symptoms including mood. It has also been beneficial for a complex case involving a young person with alcohol, cannabis, and LSD abuse, along with behavioral and psychotic symptoms. It's important to consider that lurasidone is an approved medication for treating schizophrenia in individuals as young as 13, an age group particularly susceptible to substance use. Consequently, this could make it a viable option for treatment in this younger demographic in the future
-the conclusions are unimpressive, compared to the study topic and deserve more elaboration;
Reply: Thank you for your feedback. In light of your suggestions, we plan to undertake the following revisions: We expand our concluding section to provide a more comprehensive synthesis of our key findings and insights
We changed conclusion section as follows:
In summary, the core feature of third-generation antipsychotics like aripiprazole, brexpiprazole, and cariprazine is their partial agonism at D2 dopaminergic receptors. This characteristic offers at least three potential benefits compared to the traditional antagonism approach:
In cases of mesolimbic dopaminergic overactivity linked to positive symptoms, the partial agonist competes with dopamine for receptors. This competition displaces dopamine, decreasing the system's excessive activity and returning it to a 'physiological range'.
The effects of the partial agonist don't typically include significant adverse reactions, such as extrapyramidal symptoms or hyperprolactinemia. This is because the molecule's intrinsic activity prevents a substantial reduction in dopaminergic functionality at the striatal and pituitary levels. Owing to its intrinsic activity, the partial agonist may bolster weakened dopaminergic transmission in the prefrontal cortex. This enhancement could improve cognitive dysfunctions and symptoms linked to diminished dopaminergic functionality. These pharmacodynamic properties make them especially useful in cases of substance-induced psychosis. Concerning lurasidone, its efficacy lies in its binding to D2 receptors, which helps in reducing positive symptoms. Additionally, its agonist activity at the 5HT2A and 5HT7 receptors results in the release of dopamine, contributing to the improvement of affective and cognitive symptoms. This mechanism of action could provide a foundation for additional research and the initiation of studies in patients with substance-induced psychosis.
-The notes must be revised based on the Journal's editorial rules, and at least 50% must be from the last five years. Good work!
Reply: We have reviewed all the references and increased the number of works dated within five years
Reviewer 2 Report
Comments and Suggestions for Authors
The article entitled "Third-Generation Antipsychotics and Lurasidone in the Treatment of Substance-Induced Psychosis: A Narrative Review" by Ricci and colleagues is a review article that presents experimental evidence for the use of certain antipsychotics in substance-induced psychosis.
1. The abstract could be enriched, the current form of the abstract is limited. For example, it would be useful to mention what kind of psychosis-inducing substances are discussed in the article, how these substances contribute to or induce a psychotic state, what properties the discussed drugs have, etc. Keep in mind the number of words allowed in this section according to the Guidelines for Authors.
2. It would be useful for authors to indicate in the introduction the differences between first, second and third generation antipsychotics. For example, in terms of their chemical nature and/or efficacy, better pharmacokinetic properties or safety profile, etc. It is not clear from this section how aripiprazole, cariprazine, brexpiprazole and lurasidone are relevant to the topic of the study.
3. Report the doses used for aripiprazole, cariprazine, brexpiprazole in animal models, and lurasidone in humans in Tables 1, 2, 3, and 4 of the Supplementary Appendix.
4. Please write the official nomenclature of the mentioned receptors correctly, check the following document: The Concise Guide to PHARMACOLOGY 2023/24: G Protein-Coupled Receptors (https://doi.org/10.1111/bph.16177).
5. Although the text presents valuable information, it lacks a clear and organized structure. For example, the information on aripiprazole and its effects is presented in a fragmented manner, making it difficult to follow the logical sequence of the arguments.
6. What is the meaning of SAPS and SANS (line 180)?
7. The information on line 194 is incomplete: ... "Beresford demonstrated possible (36)".
8. In the case of cariprazine and the other antipsychotics discussed, it would be very helpful if the authors provided tables with affinity values (pKi, pKa, Ka, etc.) for the different types of receptors mentioned. This would make it possible to understand the low or high selectivity of the drugs analyzed. Also, some ideas are repeated in different sections of the text, such as the description of the affinity of cariprazine for D3 receptors and its effects on the improvement of negative symptoms. Reducing redundancy would improve the clarity and conciseness of the text.
9. For lurasidone, side effects are mentioned, but the text does not provide specific details about these effects or the frequency of their occurrence. Inclusion of this information would benefit the manuscript.
10. The discussion and conclusions are very poor and should be greatly improved. I suggest that the authors avoid the structure of a research article because their manuscript is a descriptive document. For example, the methodology and results sections are unnecessary. In this way, the discussion of each of the drugs described could be expanded. It would also be positive if the authors could enrich the data already published with personal hypotheses about the effectiveness of the drugs analyzed in a comprehensive manner.
11. Some typographical errors are noted, such as the absence of spaces after certain commas and the inconsistent use of upper and lower case letters. Correcting these errors would improve the overall presentation of the text.
Comments on the Quality of English LanguageSome typographical errors are noted in the manuscript
Author Response
Dear REVIEWER 2,
The article entitled "Third-Generation Antipsychotics and Lurasidone in the Treatment of Substance-Induced Psychosis: A Narrative Review" by Ricci and colleagues is a review article that presents experimental evidence for the use of certain antipsychotics in substance-induced psychosis
Reply: We appreciate the time you have taken to evaluate our work and are grateful for the opportunity to discuss and improve upon our manuscript
1.The abstract could be enriched, the current form of the abstract is limited. For example, it would be useful to mention what kind of psychosis-inducing substances are discussed in the article, how these substances contribute to or induce a psychotic state, what properties the discussed drugs have, etc. Keep in mind the number of words allowed in this section according to the Guidelines for Authors.
Reply: Thank you for your valuable feedback regarding the abstract of our manuscript. In response to your comment, we will revise the abstract to include a concise yet informative overview of the key psychosis-inducing substances examined in our study. This will encompass a brief description of their characteristics, the mechanisms through which they contribute to or induce psychotic states, and their distinctive properties. We believe that these additions will significantly enhance the reader's understanding of the scope and focus of our research. We changed the abstract in this way:
This narrative review explores into the efficacy and tolerability of third-generation antipsychotics (TGAs) – aripiprazole, cariprazine, brexpiprazole, and lurasidone – for the management of substance-induced psychosis (SIP). SIP is a psychiatric condition triggered by substance misuse or withdrawal, characterized by unique features distinct from primary psychotic disorders. These distinctive features include a heightened prevalence of positive symptoms, such as hallucinations and delusions, in addition to a spectrum of mood and cognitive disturbances. This review comprehensively investigates various substances, such as cannabinoids, cocaine, amphetamines, and LSD, which exhibit a greater propensity for inducing psychosis.TGAs exhibit substantial promise in addressing both psychotic symptoms and issues related to substance misuse. This review elucidates the distinctive pharmacological properties of each TGA, their intricate interactions with neurotransmitters, and their potential utility in the treatment of SIP. We advocate for further research to delineate the long-term effects of TGAs in this context and underscore the necessity of adopting an integrated approach that combines pharmacological and psychological interventions. Our findings underscore the intricate and multifaceted nature of treating SIP, highlighting the potential role of TGAs within therapeutic strategies.
- It would be useful for authors to indicate in the introduction the differences between first, second and third generation antipsychotics. For example, in terms of their chemical nature and/or efficacy, better pharmacokinetic properties or safety profile, etc. It is not clear from this section how aripiprazole, cariprazine, brexpiprazole and lurasidone are relevant to the topic of the study.
Reply: Thank you for your insightful feedback on the introduction of our manuscript. Your suggestion to clarify the differences between first, second, and third generation antipsychotics is indeed valuable and will certainly enhance the comprehensiveness of our study.
In accordance with your advice, we will revise the introduction to include a detailed comparison of the three generations of antipsychotics. This comparison will focus on aspects such as their chemical nature, efficacy, pharmacokinetic properties, and safety profiles. By doing so, we aim to provide a clearer context for the relevance of these drugs in the field of psychiatry and their evolution over time. We modified the introduction as follows:
“Antipsychotic agents are established as the primary modality for treating Schizophrenia (SZ) and are generally effective in managing substance-induced psychoses as well. These medications are categorized into two groups: typical antipsychotics, also referred to as first-generation antipsychotics (FGA), and atypical antipsychotics, known as second-generation antipsychotics (SGA). The FGA class exhibits a relatively uniform pharmacological profile, whereas the SGA category displays a more diverse range, both pharmacologically and clinically.
FGAs are marked by their strong antagonism of the dopamine D2 receptor. This action is beneficial in attenuating the heightened activity of the mesolimbic dopaminergic pathway, thereby mitigating hallucinations and delusions associated with this hyperactivity. However, D2 antagonism in the mesocortical pathway can exacerbate negative and cognitive symptoms. Additionally, pronounced D2 blockade is linked to notable adverse effects, including extrapyramidal symptoms and hyperprolactinemia, arising from receptor inhibition in the nigrostriatal and tuberoinfundibular pathways. SGAs were developed with dual objectives: to reduce the incidence of these adverse effects by moderating the hyperactivity of the mesolimbic dopaminergic circuit in a more physiological manner and to address negative symptoms and cognitive deficits that were either unimproved or worsened by FGAs. Although SGAs lack a unifying pharmacological characteristic, a distinguishing feature of these drugs, in contrast to FGAs, is their concurrent antagonism of both dopamine D2 and serotonin 5-HT2A receptors. The affinity ratio for these receptors has been considered an indicative measure of their "atypicality." In recent years, the development of molecules that act as partial agonists, instead of antagonists, at dopaminergic receptors has constituted a major breakthrough in the treatment of psychosis. This innovation has led to the introduction of third-generation antipsychotics, which include medications such as cariprazine, brexpiprazole, and aripiprazole. Additionally, lurasidone, though not classified within this newer generation, demonstrates distinct pharmacodynamic properties. It exhibits antagonistic effects on D2 dopaminergic receptors and serotonin 5-HT2A and 5-HT7 receptors, and it is notable for its robust safety profile. These drugs are increasingly acknowledged for their therapeutic efficacy, safety, and ease of management in clinical settings.
3.Report the doses used for aripiprazole, cariprazine, brexpiprazole in animal models, and lurasidone in humans in Tables 1, 2, 3, and 4 of the Supplementary Appendix.
Reply: Thank you. We reported the lacking informations.
4.Please write the official nomenclature of the mentioned receptors correctly, check the following document: The Concise Guide to PHARMACOLOGY 2023/24: G Protein-Coupled Receptors (https://doi.org/10.1111/bph.16177).
Reply: Thank you. I provided to write the nomenclature according to the document
- Although the text presents valuable information, it lacks a clear and organized structure. For example, the information on aripiprazole and its effects is presented in a fragmented manner, making it difficult to follow the logical sequence of the arguments.
Reply: Thank you for your constructive feedback regarding the structure of our manuscript. We understand your concern about the organization of the text, particularly regarding the presentation of information on aripiprazole and its effects.
In response to your feedback, we will undertake a thorough revision of the manuscript with a focus on enhancing its overall structure and logical flow. We rephrase as follows:
“Aripiprazole, classified as a partial agonist within the antipsychotic drug category, exhibits lower intrinsic activity at receptors compared to full agonists (24). Its partial agonism at dopamine D2 receptors leads to 1) functional antagonism in the mesolimbic dopamine pathway, reducing positive symptoms caused by excessive dopamine activity, and 2) agonist activity in the mesocortical pathway, addressing negative symptoms and cognitive impairment due to reduced dopamine activity (25-27). This dual effect contributes to significant improvements in both positive and negative psychotic symptoms.
Moreover, aripiprazole avoids complete blockade of the nigrostriatal or tuberoinfundibular pathways, thereby preserving the avoidance of extrapyramidal symptoms and hyperprolactinemia (24). It also demonstrates a favorable cardiac safety profile, preventing QTc prolongation and causing minimal weight gain or sedation. Focusing on substance use, aripiprazole shows promise in animal models, where acute administration prevented increased locomotion induced by stimulants like amphetamine, cocaine, and methylphenidate, and attenuated their reinforcing properties without interfering with spontaneous motor activity (26). It reversed amphetamine-associated anhedonia and prevented the reinstatement of cocaine-seeking behavior (27), suggesting its potential in alleviating withdrawal symptoms linked to dopamine depletion and maintaining balanced dopamine neurotransmission in drug dependence behavior (28).
Clinical observations further support these findings. A 22-year-old male patient meeting ultra-high-risk criteria for psychosis benefited from 10 mg of aripiprazole, though without a decrease in smoking frequency (29). Another case highlighted aripiprazole's role in reducing cannabis intake in a schizophrenic patient (30). Thurston and colleagues observed that adolescents hospitalized for co-occurring psychosis and cannabis use disorder showed rapid reduction in acute psychotic symptoms with aripiprazole compared to risperidone (32). In trials comparing aripiprazole with other antipsychotics, it demonstrated superior efficacy in reducing negative symptoms of amphetamine-induced psychosis compared to risperidone, which was more effective against positive symptoms (34-35). A randomized study found aripiprazole comparable to quetiapine in alleviating psychotic symptoms but more effective in reducing cocaine dependence and usage (36). Beresford's study indicated its potential role in lowering both desire and use of cocaine in schizophrenic patients (37).
However, not all studies yielded positive outcomes. Aripiprazole was no more effective than placebo in maintaining abstinence from methamphetamine use but did facilitate treatment retention and reduce the severity of psychotic symptoms (38). “
- What is the meaning of SAPS and SANS (line 180)?
Reply: Thank you for this suggestion: we rephrase this text in this way:
“…In trials comparing aripiprazole with other antipsychotics, it demonstrated superior efficacy in reducing negative symptoms of amphetamine-induced psychosis compared to risperidone, which was more effective against positive symptoms …..”
- The information on line 194 is incomplete: ... "Beresford demonstrated possible (36)".
Reply: I apologize for this oversight.”Beresford's study indicated its potential role in lowering both desire and use of cocaine in schizophrenic patients (37).”
8. In the case of cariprazine and the other antipsychotics discussed, it would be very helpful if the authors provided tables with affinity values (pKi, pKa, Ka, etc.) for the different types of receptors mentioned. This would make it possible to understand the low or high selectivity of the drugs analyzed. Also, some ideas are repeated in different sections of the text, such as the description of the affinity of cariprazine for D3 receptors and its effects on the improvement of negative symptoms. Reducing redundancy would improve the clarity and conciseness of the text.
Reply: Thank you for your invaluable suggestions regarding the inclusion of affinity values for antipsychotics and the reduction of redundancy in our manuscript. In accordance with your suggestion, we will incorporate tables that detail the affinity values of the antipsychotics discussed in our study. Wh hope that These tables will present the data in a clear, concise manner, allowing readers to easily comprehend the selectivity and potency of these drugs at various receptors. You can find this table in supplementary data, as Table 6.
Furthermore, we acknowledge the issue of redundancy in our text, particularly concerning the description of cariprazine’s affinity for D3 receptors and its effects on negative symptoms. We changed the tex as follows
“Cariprazine, a third-generation antipsychotic drug approved for schizophrenia (39), is notable for its partial agonism towards D2 receptors. Distinguishing itself from other Second-Generation Antipsychotics (SGAs), it demonstrates a lower intrinsic activity at the D2 receptor compared to aripiprazole and brexpiprazole, approximately 0.15. This positioning suggests a minimized capacity to activate the D2 receptor, offering an intermediary profile between aripiprazole and traditional receptor antagonists with zero intrinsic activity (39-42). The reduced intrinsic activity of cariprazine is associated with a decreased likelihood of side effects such as restlessness and akathisia, commonly observed with aripiprazole (40).
Cariprazine's primary distinction lies in its high affinity for the dopaminergic D3 receptor, exceeding that of endogenous dopamine. This high D3 affinity is significant, considering that most antipsychotics have a lower affinity for this receptor, leading to inadequate D3 receptor occupancy in the brain. This unique affinity of cariprazine is crucial for its clinical efficacy, as indicated by PET studies in schizophrenic patients, demonstrating a stronger effect on the D3 receptor compared to the D2 receptor (41-47, 49).
Cariprazine also exhibits significant serotonergic activity, with high affinity for the 5-HT2B receptor and moderate affinity for the 5-HT2A and 5-HT1A receptors. This receptor profile is somewhat distinct from many SGAs, which tend to have high affinity for the 5-HT2A receptor. The 5-HT2A/D2 affinity ratio of cariprazine is lower than that of other antipsychotic drugs. Additionally, it has low affinity for the 5-HT7 and 5-HT2C receptors, as well as the noradrenaline α1A and α1C receptors. Its interactions with the 5-HT6, α1a, α2b receptors, and other potential targets are minimal in therapeutic activity (42-43).
When comparing cariprazine with other partial agonists, such as blonanserin (48), an antagonist of dopaminergic D2 and D3 receptors, its D3/D2 affinity ratio is notably higher. This suggests a potential influence on negative symptoms, a theory supported by its unique receptor action. In contrast, brexpiprazole shows higher 5-HT2A/D2 and 5-HT1A/D2 ratios. The combined dopaminergic and serotonergic receptor activities of these drugs modulate specific brain circuits and neurotransmitter release, as observed in increased dopamine release in the prefrontal cortex with olanzapine and lurasidone, enhancing cognitive functions (50). Similarly, cariprazine affects neurotransmitter release in the nucleus accumbens and hippocampus, mainly via D3 receptor interaction, influencing levels of dopamine, norepinephrine, serotonin, and glutamate. This effect is comparable to that of dopaminergic D3 antagonists, contributing to cariprazine's therapeutic impact (50).
In conclusion, the receptor profile of cariprazine provides significant insights into its mechanism of action and its ability to modulate various neurotransmitter systems. Unlike other antipsychotics, cariprazine's efficacy is not only based on its receptor profile but also on its capacity to modulate intracellular mechanisms downstream of these receptors (51-52). Its dual action as a partial agonist at both D2 and D3 receptors, with a stronger effect on the latter, is particularly effective in improving negative symptoms, setting it apart from other SGAs (53)..”
- For lurasidone, side effects are mentioned, but the text does not provide specific details about these effects or the frequency of their occurrence. Inclusion of this information would benefit the manuscript.
Reply: Thank you for pointing out the need for more detailed information on the side effects of lurasidone in our manuscript. We agree that including specifics about these effects and their frequency of occurrence would significantly enhance the comprehensiveness and utility of our work. We added this tex with specific references
“. Lurasidone demonstrated good tolerability, with consistent side effects observed in both short-term and long-term use. In trials spanning six weeks, the most frequently reported adverse reactions to lurasidone included drowsiness, restlessness, nausea, Parkinson-like symptoms, and sleeplessness (83). Lurasidone demonstrated good tolerability, with consistent side effects observed in both short-term and long-term use. In trials spanning six weeks, the most frequently reported adverse reactions to lurasidone included drowsiness, restlessness, nausea, Parkinson-like symptoms, and sleeplessness (83). Finally, Lurasidone was as- sociated with lower weight gain and metabolic disturbances than brexpiprazole (84-85).”
- The discussion and conclusions are very poor and should be greatly improved. I suggest that the authors avoid the structure of a research article because their manuscript is a descriptive document. For example, the methodology and results sections are unnecessary. In this way, the discussion of each of the drugs described could be expanded. It would also be positive if the authors could enrich the data already published with personal hypotheses about the effectiveness of the drugs analyzed in a comprehensive manner.
Reply: I am infinitely grateful for this suggestion. In line with the editorial guidelines and the advice of the first reviewer as well, we have expanded the discussions and the conclusion, both in going more into the specifics of the drugs in question and in drawing personal conclusions. We have changed the text as follows:
From the above-mentioned data, third-generation antipsychotics and lurasidone emerge as a promising therapeutic strategy in the treatment of substance-induced psychoses. Aripiprazole has been effective in improving a wide range of psychotic symptoms, including both positive and negative aspects, as well as impacting substance use disorders positively. Its mode of action is unique; it's a partial agonist, meaning it doesn't stimulate receptors as strongly as full agonists. Its effectiveness lies in its dual role: it diminishes positive symptoms by antagonizing the mesolimbic dopamine pathway and improves negative symptoms and cognitive deficits by activating the mesocortical pathway. This selective mechanism helps aripiprazole to avoid severe side effects like motor disorders and elevated prolactin levels, common in other antipsychotics. Additionally, it's known for its cardiac safety, causing negligible QTc prolongation and having a low risk of weight gain or sedation. In research with animals, aripiprazole has been observed to curb the heightened activity caused by stimulants such as amphetamine, cocaine, and methylphenidate, and reduce their addictive qualities without hampering normal motor functions. It also reverses the lack of pleasure associated with amphetamine use and hinders the recurrence of cocaine-seeking behaviors. These findings indicate that aripiprazole might be effective in easing withdrawal symptoms associated with dopamine deficiency and, due to its broad receptor activity, could represent a new strategy for achieving balanced dopamine levels in the treatment of drug addiction.
Cariprazine’s receptor profile, particularly its high affinity for the D3 receptor and reduced intrinsic activity at D2 receptors, makes it an effective treatment for schizophrenia, improving both positive and negative symptoms. Its ability to modulate different neurotransmitter systems further highlights its potential as a distinct and effective antipsychotic medication. Cariprazine's role in substance abuse treatment is noteworthy. From the studies reviewed, emerge its effectiveness in reducing the stimulating effects of substances like cocaine and mitigating cravings and relapses. This effect is possibly due to its partial agonism at D2 and D3 receptors. Case studies show cariprazine’s beneficial impact on schizophrenic patients with a history of substance abuse and its effectiveness in treating psychosis induced by substances like methamphetamine.
Brexpiprazole exhibits potential efficacy in the domain of substance abuse therapy. Primarily indicated for the management of schizophrenia and as an adjunctive treatment in Major Depressive Disorder (MDD), its unique pharmacodynamic properties extend to the mitigation of substance-induced psychotic disorders. Operating as a partial agonist at the 5-HT1A and D2 neuroreceptors, brexpiprazole also engages with noradrenergic receptors. Consequently, albeit preliminary and limited in scope, research indicates its utility in addressing psychotic sequelae associated with cannabis consumption and in modulating dopaminergic activity in heroin-exposed rodents.
Lurasidone has shown potential in treating psychopathological conditions related to substance abuse, although research in this area is limited. Known for its antagonistic action at dopamine D2 and serotonin 5-HT2A receptors, and strong affinity for serotonin 5-HT7 receptors, lurasidone is unique in its partial agonism at 5-HT1A receptors. This receptor profile contributes to its low sedative effects and minimal impact on weight and cognitive functions, making it an appealing option in treating substance-induced psychoses. In the context of substance abuse recent studies has shown lurasidone to be effective in treating young individuals with substance-induced psychosis, particularly from cannabis, improving various symptoms including mood. It has also been beneficial for a complex case involving a young person with alcohol, cannabis, and LSD abuse, along with behavioral and psychotic symptoms. It's important to consider that lurasidone is an approved medication for treating schizophrenia in individuals as young as 13, an age group particularly susceptible to substance use. Consequently, this could make it a viable option for treatment in this younger demographic in the future.
Finally, despite aripiprazole showing more scientific evidence, there are currently limited specific clinical studies demonstrating its long-term efficacy. Several critical issues emerged from our observation, including:
Firstly, these drugs may exhibit variable efficacy in managing psychotic symptoms associated with substance use, as the nature and complexity of substance-induced psychoses can differ significantly from those purely psychiatric in origin. Additionally, third-generation antipsychotics may not optimally address specific aspects of substance-induced psychoses, such as the management of cognitive disorders and compulsive impulses, which often characterize these conditions.
Another significant limitation concerns side effects. Third-generation antipsychotics, although to a lesser extent, can induce metabolic disorders and increase the risk of tardive dyskinesia, posing additional challenges in the treatment of substance-induced psychoses. Furthermore, patient compliance may become an issue due to these side effects, compromising treatment adherence.
- Conclusions
In summary, the core feature of third-generation antipsychotics like aripiprazole, brexpiprazole, and cariprazine is their partial agonism at D2 dopaminergic receptors. This characteristic offers at least three potential benefits compared to the traditional antagonism approach: In cases of mesolimbic dopaminergic overactivity linked to positive symptoms, the partial agonist competes with dopamine for receptors. This competition displaces dopamine, decreasing the system's excessive activity and returning it to a 'physiological range'. The effects of the partial agonist don't typically include significant adverse reactions, suchcs extrapyramidal symptoms or hyperprolactinemia. This is because the molecule's intrinsic activity prevents a substantial reduction in dopaminergic functionality at the striatal and pituitary levels. Owing to its intrinsic activity, the partial agonist may bolster weakened dopaminergic transmission in the prefrontal cortex. This enhancement could improve cognitive dysfunctions and symptoms linked to diminished dopaminergic functionality. These pharmacodynamic properties make them especially useful in cases of substance-induced psychosis.
Concerning lurasidone, its efficacy lies in its binding to D2 receptors, which helps in reducing positive symptoms. Additionally, its agonist activity at the 5HT2A and 5HT7 receptors results in the release of dopamine, contributing to the improvement of affective and cognitive symptoms. This mechanism of action could provide a foundation for additional research and the initiation of studies in patients with substance-induced psychosis.
Finally, it is essential to consider that third-generation antipsychotics may not fully address the underlying cause of substance-induced psychoses, linked to the intake of specific psychoactive substances. Therefore, a comprehensive therapeutic approach should include not only symptomatic treatment with antipsychotics but also targeted intervention for substance dependence and relapse prevention through psychological interventions and rehabilitation programs. In summary, while third-generation antipsychotics are valuable in treating psychoses, their effectiveness and appropriateness in substance-induced psychoses require particular attention and an integrated approach.
- Some typographical errors are noted, such as the absence of spaces after certain commas and the inconsistent use of upper and lower case letters. Correcting these errors would improve the overall presentation of the text.
Reply: Thank you for highlighting the typographical errors in our manuscript. Undertake a meticulous review of the entire manuscript to identify and correct all typographical errors, including the absence of spaces after commas and inconsistencies in the use of upper and lower case letters.
Utilize advanced proofreading tools in addition to manual checks to ensure that we capture and rectify all such errors.
Implement a final review process involving multiple team members to ensure thoroughness and accuracy.
Round 2
Reviewer 1 Report
Comments and Suggestions for Authors
Dear authors, Excellent work from every point of view. I am satisfied with the implementations made and suggest publication. Have a good day
Reviewer 2 Report
Comments and Suggestions for Authors
No further comments